# Comparative efficacy of eight therapeutic methods in the treatment of left main coronary artery disease: a Bayesian network meta-analysis protocol

Biao Hou,[1] Manlin Chen,[2] Qin Li,[1] Weimin Huang ![ORCID],[1] Liang Wang[1,3]

[1]Inner Mongolia Medical University, Hohhot, China
[2]Bazhong Central Hospital, Sichuan, China
[3]Cardiothoracic surgery department, Inner Mongolia Baotou City Central Hospital, Baotou, Inner Mongolia, China

**Correspondence to**
Dr Weimin Huang;
huang_manlin2021@163.com

## ABSTRACT

**Introduction** As for coronary artery bypass grafting, although there are many direct comparative studies on different minimally invasive methods and traditional thoracotomy (off-pump/on-pump), there is still a lack of further ranking and summary of the efficacy of all surgical methods for left main coronary artery (LMCA) lesions. Combined with the current controversial views, this study aims to introduce a planned network meta-analysis (NMA) in detail, with a view to comparing the long-term efficacy and safety of multiple therapeutic methods in the treatment of patients with LMCA disease, and finally providing some reference bases for the best selection of clinical schemes.

**Method and analysis** PubMed, Embase, Web of Science and The Cochrane Library databases will be collected from inception to June 2022 to compare the efficacy of different surgical methods in randomised controlled trials (RCTs) for LMCA disease. Main outcome endpoints: major adverse cardiovascular events, including mortality, myocardial infarction, stroke and revascularisation. Secondary outcome endpoints: (1) operation-related time, (2) the amount of blood transfusion, (3) complications including secondary thoracotomy, postoperative new atrial fibrillation, wound infection, (4) physiological score and psychological score, (5) time return to work and (6) total hospitalisation costs. The methodological quality of included RCTs will be assessed according to the Cochrane bias risk table. The Bayesian NMA will be conducted by STATA V.16.0.

**Ethics and dissemination** The essence of this study is to summarise and analyse the original data without the approval of the ethics committee. Our research does not involve ethical issues, and the results will be published in peer-review journals.

**PROSPERO registration number** CRD42021274712.

## STRENGTHS AND LIMITATIONS OF THIS STUDY

⇒ This is the one Bayesian network analysis to comprehensively compare various therapeutic methods of left main coronary artery disease.
⇒ The retrieval time is long, the scope is wide and the quality of all included articles will be strictly evaluated by two members with evidence-based medicine experience independently according to the manual.
⇒ There may be mixed factors, such as different surgeon experience and population baseline characteristics, but comprehensive analysis methods such as subgroup analysis and the stability results of large samples may conceal this effect.

has been the gold standard for the treatment of LMCA diseases.[2] PCI was only used as a substitute for high-risk patients or not suitable for surgical patients.[3] There were some randomised controlled trials (RCTs) on PCI and CABG,[1 4 5] but the results showed some contradictory therapeutic outcomes.

In order to reduce the complications caused by extracorporeal circulation technology, off-pump CABG (OPCAB) has been carried out, and the relevant study[6] has shown that OPCAB can significantly reduce mortality and morbidity. However, some claim that OPCAB cannot provide the benefits of complete revascularisation.[7 8] Others sought a compromise between the two surgeries, namely mini cardiopulmonary bypass coronary artery bypass (MECC), and there was a network meta-analysis[9] reported RCTs of this approach.

With the development of medical technology, other surgical methods for the treatment of coronary heart disease include: minimally invasive under direct vision (MIDCAB),[10] robot-assisted coronary artery bypass grafting (RECAB),[11] total endoscopic coronary artery bypass grafting (TECAB)[12] and hybrid coronary artery revascularisation (HCR),[13] etc.

## INTRODUCTION

Left main coronary artery (LMCA) stenosis would involve large areas of myocardium and increase the risk of major adverse cardiac events.[1] LMCA treatment strategies include the coronary artery bypass grafting (CABG) and the percutaneous coronary intervention (PCI). For more than 40 years, conventional extracorporeal circulation coronary (CECC)

The different anatomical approaches of direct-viewing minimally invasive surgery may make surgeons feel stranger, and there are drawbacks that the assistants' vision is incomplete and unable to cooperate with them.[10] Similarly, RECAB and TECAB technologies both need higher technical threshold requirements and longer learning curve. If the key process of operation is not smooth, the above methods are likely to be converted to the sternotomy approach.[11] For HCR, first of all, the sequence of PCI and CABG is currently controversial[14]; second, the cost of hybrid technology is high, which is difficult for patients to accept and the promotion is limited.[15]

Thus, under different circumstances, the best strategy for revascularisation of left main lesions is still controversial. The purpose of this study is to summarise the above surgery methods for coronary heart disease, compare and rank them by using mesh meta-analysis, so as to provide some decision-making help for clinicians.

## METHODS AND ANALYSIS
### Literature search
The protocol was formulated according to the 2015 checklist of the Preferred Reporting Items for Systematic Reviews and Meta-Analyses Protocols (PRISMA-P).[16 17] The actual study will be implemented according to the PRISMA statement[18] and research guideline.[19]

Two authors (WH and BH) will independently collect and screen RCTs on different surgical methods (including PCI) for the treatment of coronary heart disease from PubMed, Embase, Web of Science and The Cochrane Library databases. The search time limit is from the establishment of the database to June 2022.

The retrieval will be performed using a combination of grid words and free text words. Some English terms are 'Coronary Disease, Left Main Disease, Coronary Artery Bypass, Myocardial Revascularization, CABG, Surgical Procedures, Percutaneous Coronary Intervention, Robotic Surgical Procedures, Video-Assisted Surgery, Thoracoscopes, Hybrid, Thoracotomy'. The detailed search strategy is described in online supplemental file 1.

### Eligibility criteria
Studies will be selected according to the PICO criteria: Patients (P), Intervention (I), Comparators(C), and Outcome(s) of interest (O).

Patients(P) : All patients who were included in the study have undergone CABG or PCI for the first time. The only difference in population characteristics should be different treatment methods of coronary heart disease. RCTs will be included in this meta-analysis.

Intervention(I): The methods should include CECC, MECC, OPCAB, MIDCAB, RECAB, TECAB, HCR and PCI.

Comparators(C): The on-pump coronary artery bypass (ONCAB or CECC) operation method will be performed through median thoracotomy, which always used to the control to compare the main outcomes such as common postoperative complications, adverse cardiovascular and cerebrovascular events.

Outcomes(O): Primary outcomes: Major adverse cardiovascular event endpoints should include the numbers of mortality, myocardial infarction, stroke, revascularisation during the follow-up time which is at least 1 year. The occurrence of adverse events can be counted respectively during their hospitalisation, 6 months after operation, 1 year or more after operation. Secondary outcomes: (1) surgery-related time, (2) the amount of blood transfusion, (3) complications (4) physical score and psychological score, (5) time return to work and (6) total hospital costs. (Special definition: 'blood transfusion': it should include the amount of blood transfusion during the operation and during the stay in the cardiac surgical intensive care unit. It often refers to the cumulative amount of blood transfusion during the hospitalisation. 'complications' here refer to postoperative wound infection, pneumonia, liver and kidney dysfunction, new postoperative atrial fibrillation, etc. 'physiological score and psychological score' are the scores determined by some literatures according to short form health survey 12 and 36 (SF-12 and SF-36) quality of life questionnaire. The higher the score, the better the curative effect).

Qualification criteria have been determined by two researchers (WH and BH), and then discussed and agreed with other authors (QL, MC and LW). As follows:

Inclusion criteria: (1) RCT trials; (2) All patients involved in the study were treated with CABG or PCI for the first time.

Exclusion criteria: (1) non-English literature; (2) patients with other major diseases that may affect the surgical efficacy (such as severe pulmonary hypertension); (3) unreasonable research design; (4) the full text or outcome indicators less than 3; (5) repeated publications by the same institution or author; (6) continuity variables are not represented by mean±SD.

### Selection process
First, two authors (WH and BH) will independently use the EndNote V.X9 software to classify and organise the searched literature according to surgical methods. Second, the excluded documents will be to place in a separate folder and marked to explain why they are excluded. The third step, by reading the titles and abstracts included in the literature, we would note the surgical grouping comparison (such as OPCAB vs MIDCAB) for future verification. The fourth step, by reading the full text, again exclude irrelevant literature, classify and mark. The fifth step is to judge by the third party (MC or QL) if there is disagreement. We will strictly follow the above steps to ensure the high-quality and the comprehensiveness of the included literature.

### Data extraction
The person responsible for screening (WH and BH) will be asked to be familiar with the data in advance, and the data extraction table would be improved according to the

situation, and the scoping studies will be conducted as recommended.[20]

The extracted data will include the publication years of the study, institutional background, random methods, baseline characteristics of patients (age, gender, the body mass index, the SYNTAX score, concomitant diseases and the number of revascularised vessels), various outcome endpoints, missing visits and statistical methods. In addition, we will collect data on the type of surgery (elective, urgent or emergency), surgical indications (acute vs chronic coronary syndrome) and the medical therapy patients (antiplatelet therapy) during the perioperative period, as appropriate. In case of lack of data, we will contact the author by email.

### Risk of bias in individual studies

Two reviewers (WH and MC) will be assessed. Any differences between reviewers will be resolved by discussing or requiring a third reviewer (BH) to assess. The included RCTs were independently assessed according to the Cochrane Handbook for Systematic Reviewers bias risk assessment criteria.[18] Each study will be graded by scores, as follows: A (low risk): >7 stars, B (medium risk): 5–7 stars and C (high risk): <5 stars.[21]

### Subgroup analysis

The heterogeneity may come from such factors as large differences in the years of publication, different population backgrounds and inconsistent acceptance criteria for patients. First of all, we will preliminarily evaluate the reliability of the meta-analysis results through sensitivity analysis (excluding some low-quality studies). Then, on this basis, we will also conduct subgroup analysis to compare the efficacy of each subgroup, so as to determine whether different regions, races and other factors may affect the research results.

### Statistical analyses

In previous Meta-analysis publication,[22] we used ADDIS software, it will be different next. We plan to use STATA V.16.0 software to draw a network diagram of the comparison of various interventions, and use Markov Chain Monte Carlo method to simulate, the number of iterations is set to 50,000.[23] Interstudy heterogeneity will be evaluated by the Q statistic, where $p<0.10$ will be considered statistically significant and informative by $I^2$ statistic, where $I^2 \geq 50\%$ will indicate heterogeneity. We will perform subgroup meta-analysis to assess differences.[24] In order to evaluate whether publication bias exists in the whole network, this study intends to use comparison-correction funnel plot.[25] The league table will be calculated for each main outcome endpoint, and the ranking results are reflected by the area under the cumulative ranking curve.[26] To sum up, we will use the following two kinds of software for analysis at the same time, and the whole process will be checked by statistical experts. The general steps are as follows: first, we will make a network diagram and some forest diagrams according to the preprocessed

data, and then, we will draw some ranking charts (net-league tables) for the efficacy comparison of each treatment method in strict accordance with the operating specifications. For each endpoint that meets the inconsistency test model, we will actively look for the source of heterogeneity, and conduct sensitivity analysis and subgroup analysis, eventually give a reasonable explanation for the results. Finally, we would cumulative probability of all observed endpoints and rank these treatments from priority to inferiority in tabular forms.

The software to be used in this study are STATA V.16.0 (Stata) and Review Manager V.5.4 (Oracle, The Cochrane Collaboration, 2020).

### DISCUSSION

As mentioned above, although the ONCAB or CECC has always been the gold standard for the treatment of LMCA diseases, with the rise of minimally invasive surgery, the discussion about the best strategy for revascularisation of left main artery lesions is controversial in clinic.[2] Although numerous RCTs have compared CABG with PCI, no studies have been powered to detect a difference in mortality during the long follow-up among them. One study[27] has reported that no benefit for CABG over PCI was seen in patients with left main disease (CABG had a mortality benefit over PCI in patients with multivessel disease, and those with diabetes and higher coronary complexity.).

Although the current clinical guidelines have pointed that the SYNTAX score could help select the vascular reconstruction strategy for unprotected left main disease,[28] one study of 10-year outcomes has shown that the discriminative capacity of SYNTAX score was relevant in the PCI group but not in the CABG group.[29]

Previous meta-analysis[9–13] showed that compared with traditional CABG, different surgical methods had certain advantages in different indicators. However, for the newly developed surgical treatment methods in recent years, such as robotic CABG,[11] the number of RCTs is limited and lacks convincing, and there is no systematic and comprehensive comparison.

In order to ensure the quality of research, the authors will follow strict guidelines in the review process and their reports, such as PRISMA-P and PRISMA (Online supplemental file 2)for Scoping Reviews.[30] In order to avoid possible methodological defects, we will use the latest guideline provided by the Joanna Briggs Institute in 2020 when conducting the scope review.[31] Our proposed programme was registered in a predefined manner to increase the transparency and reliability of the review results.[32]

Of course, our research also has limitations. For example, although there are extensive search strategies, we only include literature with English language. Others may worry that there are confounding factors, such as different surgeons experience, population baseline characteristics, which may cause the different results of the

entire study. However, as long as enough randomised controlled studies that meet the eligibility criteria are included, the stable results of the network analysis of large samples will mask this effect. In addition, when indirect comparison cannot be conducted in any case, we will conduct reliable direct comparison analysis results. If quantitative synthesis is not appropriate, narrative synthesis will be used.

In summary, the study planned by our team may be a relatively comprehensive and authentic comparison in the treatments about LMCA disease. The analysis results will be used to provide some decision-making help for the best choice of which CABG strategy or PCI.

## Patient and public involvement

As the proposed systematic review will be conducted based on published studies, no patients and members of the public will be directly involved.

## Amendments

Any amendments to this protocol will be documented.

## Planned start and end date

The review is planned to start on 1 November 2021 and end on 1 June 2023.

## Ethics and dissemination

The essence of this study is to summarise and analyse the original data without the approval of the ethics committee. Our research does not involve ethical issues, and the results will be published in peer-review journals.

**Contributors** WH: concept research methodology, database search, article screening, data extraction, quality evaluation and drafting. BH: database search, article screening and data extraction will be conducted. QL: make the screening form and judge the inconsistent opinions. MC: Literature quality evaluation and statistical analysis. LW: participate in the outcome discussion. All authors will read and approve the final manuscript.

**Funding** The authors have not declared a specific grant for this research from any funding agency in the public, commercial or not-for-profit sectors.

**Competing interests** None declared.

**Patient and public involvement** Patients and/or the public were not involved in the design, or conduct, or reporting, or dissemination plans of this research.

**Patient consent for publication** Not applicable.

**Provenance and peer review** Not commissioned; externally peer reviewed.

**ORCID iD**
Weimin Huang http://orcid.org/0000-0003-2556-6214

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
