## [Reviewer comments · BMJ Open]

ARTICLE DETAILS

TITLE (PROVISIONAL)	Comparative efficacy of 8 therapeutic methods in the treatment of left main coronary artery disease: a Bayesian network meta-analysis protocol.
AUTHORS	Hou, Biao; Chen, Manlin; Li, Qin; Huang, Weimin; Wang, Liang

VERSION 1 – REVIEW

REVIEWER	Bularga, Anda University of Edinburgh Division of Clinical and Surgical Sciences
REVIEW RETURNED	24-Jan-2022

GENERAL COMMENTS	1. The authors will be conducting a network meta-analysis evaluating different surgical techniques for revascularisation of the left main coronary artery – the title of the study and the abstract could clarify this further by replacing “coronary heart disease” with “left main coronary artery” as this is more precise description of the proposed meta-analysis. 2. The authors summarise the data extraction variables on page 4. Are the authors interested to collect data on the type of surgery (elective, urgent or emergency), indication for surgery (acute vs chronic coronary syndrome) and the medical therapy patients receive at the time of surgery and after (particularly antiplatelet therapy)? 3. Are the authors planning to perform subgroup meta-analysis and are they able to specify subgroups of interest? 4. It would be advised that the authors include further information on the ‘Statistical analysis’ paragraph of the study protocol – explaining how will the network meta-analysis be performed particularly how will the treatment effect estimates be obtained? Minor comments: 1. Please ensure that all abbreviations are explained where first encountered. 2. Please add in “heterogeneity”, (page 4, line 16) after ‘Interstudy’. 3. It is unclear what do the authors mean by “number of blood revascularization” on page 4, line 3 – is this the number of revascularized vessels.
---

	4. One of the specified secondary outcomes is “transfusion’ – can the authors clarify timing of transfusion is this during operation, immediately post op or for longer duration? 5. Duration of follow up – what will be the timing for MACE?
--	---

REVIEWER	Li, Hui-biao Guangzhou University of Traditional Chinese Medicine First Affiliated Hospital, Department of pharmacy
REVIEW RETURNED	24-Jan-2022

GENERAL COMMENTS	1.
----

REVIEWER	Nalluri, Nikhil Staten Island University Hospital
REVIEW RETURNED	31-May-2022

GENERAL COMMENTS	I commend you for taking a very interesting and relevant topic. The objectives and methodology was clearly explained. I am interested to see the results of your final metanalysis would encourage you to look into the limitations as well
---

VERSION 1 – AUTHOR RESPONSE

To the Reviewers:

1. Question: The authors will be conducting a network meta-analysis evaluating different surgical techniques for revascularisation of the left main coronary artery – the title of the study and the abstract could clarify this further by replacing “coronary heart disease” with “left main coronary artery” as this is more precise description of the proposed meta-analysis.

Response: The authors very much agree with this view, and have replaced "coronary heart disease" with "left main coronary artery"(in the title and abstract), making the main idea of the protocol more clearly.

2. Question: The authors summarise the data extraction variables on page 4. Are the authors interested to collect data on the type of surgery (elective, urgent or emergency), indication for surgery (acute vs chronic coronary syndrome) and the medical therapy patients receive at the time of surgery and after (particularly antiplatelet therapy)?

Response: As required, the authors have made a detailed explanation in the column of data extraction variables, and added the collection of indicators such as operation type, indication for surgery and the medical therapy patients (particularly antiplatelet therapy) (page 4, line 13-16).

3. Question: Are the authors planning to perform subgroup meta-analysis and are they able to specify subgroups of interest?

Response: The authors will conduct sensitivity analysis and subgroup analysis according to the actual situation, such as grouping according to different regions and different publication years of literature. Please see lines 26-31 on page 4 for details.

4. Question: It would be advised that the authors include further information on the ‘Statistical analysis’ paragraph of the study protocol – explaining how will the network meta-analysis be performed particularly how will the treatment effect estimates be obtained?

Response: The authors have added more information in the "statistical analysis" section of the study protocol to explain how to conduct a network meta-analysis, especially how to evaluate the treatment effect, such as comparing the SUCRA value, ranking the probabilities of observation endpoints by using the list method, and comprehensively judging the safety and effectiveness of one treatment method. Please see lines 42-46 on page 4 for details.

To the Reviewer's minor comments:

Question: 1. Please ensure that all abbreviations are explained where first encountered.

Response: We have checked to ensure that all abbreviations are explained where they are first encountered. (What needs to be explained is, the SYNTAX score of the page 5 is a professional vocabulary in the cardiovascular field, and is directly cited in major literature and guidelines.)

Question: 2. Please add in "heterogeneity", (page 4, line 16) after 'Interstudy'.

Response: OK, we have added "heterogeneity" after "interstudy" (page 4, line 37).

Question: 3. It is unclear what do the authors mean by "number of blood revascularization" on page 4, line 3 – is this the number of revascularized vessels.

Response: Yes, this refers to the number of revascularized vessels, which we have described completely - Page 4, line 13.

Question: 4. One of the specified secondary outcomes is "transfusion" – can the authors clarify timing of transfusion is this during operation, immediately post op or for longer duration?

Response: The "blood transfusion": it should include the amount of blood transfusion during the operation and during the stay in the Cardiac Surgical Intensive Care Unit (CSICU). It often refers to the cumulative amount of blood transfusion during the hospitalization. —page 3, line 27-29.

Question: 5. Duration of follow up – what will be the timing for MACE?

Response: After discussion, we believe that the follow-up time included in the study should be at least 1 year. Therefore, we believe that the time points of MACE events should be during hospitalization, 6 months after operation and 1 year after operation—page 3, line 23-25.

We will continue to wait for your good reply on the manuscript decision.